# An ultrasmall organic synapse for neuromorphic computing

Shuzhi Liu [1,2,8], Jianmin Zeng[1,8], Zhixin Wu[1,8], Han Hu[3,8], Ao Xu[4], Xiaohe Huang[5], Weilin Chen[1], Qilai Chen[6], Zhe Yu[6], Yinyu Zhao[3], Rong Wang[3], Tingting Han[4], Chao Li[4], Pingqi Gao[6], Hyunwoo Kim[7], Seung Jae Baik[7], Ruoyu Zhang [3] ✉, Zhang Zhang [4] ✉, Peng Zhou [5] ✉ & Gang Liu [1] ✉

High-performance organic neuromorphic devices with miniaturized device size and computing capability are essential elements for developing brain-inspired humanoid intelligence technique. However, due to the structural inhomogeneity of most organic materials, downscaling of such devices to nanoscale and their high-density integration into compact matrices with reliable device performance remain challenging at the moment. Herein, based on the design of a semicrystalline polymer PBFCL$_{10}$ with ordered structure to regulate dense and uniform formation of conductive nanofilaments, we realize an organic synapse with the smallest device dimension of 50 nm and highest integration size of 1 Kb reported thus far. The as-fabricated PBFCL$_{10}$ synapses can switch between 32 conductance states linearly with a high cycle-to-cycle uniformity of 98.89% and device-to-device uniformity of 99.71%, which are the best results of organic devices. A mixed-signal neuromorphic hardware system based on the organic neuromatrix and FPGA controller is implemented to execute spiking-plasticity-related algorithm for decision-making tasks.

Electronic gadgets that are capable of thinking, making decisions, and interacting with human beings are dream targets of artificial intelligence techniques[1–4]. In biological neural systems, manipulation of the physiological signals is achieved by cross-membrane interactions between biomolecules, such as neurotransmitters, and various metal ion species in neurons and synapses. Emulating the molecular level superior computing power of biological systems with advanced semiconductor devices and microelectronic chips is an inevitable course of developing such brain-inspired intelligence[5,6]. Beneficial from the inherent nanometer-scale material properties that endow versatile electrical characteristics for biomimicking signal processing[7–9], as well as intrinsic biocompatibility and mechanical softness[10–12], organic and nanomaterial-based synapses are essential building blocks of implementing neuromorphic computing schemes for future human-machine fusion intelligence[13–16].

To achieve effective neuromorphic computing with organic synaptic devices, great efforts have been made to establish appropriate working mechanisms, *viz*., charge transfer, redox transition, structure reconfiguration, and ion migration, to control the electronic and electrical status of single molecules and their assemblies[17–24]. Developing novel organic materials as the physical carrier of neuromorphic computing schemes is another critical requirement to render high-density integration of miniaturized devices to resemble the complex biological neural network of neurons and synapses with

[1]Department of Micro/Nano Electronics, School of Electronic Information and Electrical Engineering, Shanghai Jiao Tong University, Shanghai 200240, China. [2]School of Chemistry and Chemical Engineering, Shanghai Jiao Tong University, Shanghai 200240, China. [3]Ningbo Institute of Materials Technology and Engineering, Chinese Academy of Sciences, Ningbo 315201, China. [4]School of Microelectronics, Hefei University of Technology, Hefei 230601, China. [5]State Key Laboratory of ASIC and Systems, School of Microelectronics, Fudan University, Shanghai 200433, China. [6]School of Materials, Sun Yat-Sen University, Guangzhou, Guangdong 510275, China. [7]School of Electronic and Electrical Engineering, Hankyong National University, Anseong-si, Gyeonggi-do 17579, Korea. [8]These authors contributed equally: Shuzhi Liu, Jianmin Zeng, Zhixin Wu, Han Hu. ✉e-mail: zhangruoy@nimte.ac.cn; zhangzhang@hfut.edu.cn; pengzhou@fudan.edu.cn; gang.liu@sjtu.edu.cn

hyperparallel processing capability. Combining the *in-operando* spectroscopy and scanning probe microscopy measurements, Venkatesan and Williams correlate the molecular redox transition with switching characteristics in micrometer organic memristors and demonstrate the possibility of constructing nanoscale organic synaptic devices[11,25,26]. Molecular structure reconfiguration-based single-molecule neuromorphic devices are also reported[27,28]. Note that these demonstrations are realized on either material aspects or individual cells that are far from the interconnected neuromorphic devices desired in real chips. In order to realize scalable integration of devices with considerable size and interconnectivity for functionality applications, researchers also tried to optimize lithography techniques to shrink the electrode linewidth of organic synapses to a sub-micrometer range[29–32]. Nevertheless, due to the structure inhomogeneity of most organic materials that are associated with the presence of highly asymmetric chromophores and lack of strong intermolecular forces to achieve long-range ordering, the electronic procedures (e.g., generation, transport, and collection of charge carriers) occurred in organic neuromorphic devices are usually spatially uneven. Not to mention the molecular scale computing scheme, downscaling of such neuromorphic devices to the nanometer range with reliable performance is still difficult. Therefore, it deserves in-depth investigation to unearth the structure-property relationship between the molecular arrangements, multiscale ordering, and electronic events in organic material to implement ultrasmall synapses for high-performance neuromorphic computing.

High-performance solid-state electronic materials usually show a tendency to arrange (at least partially) orderly to impart optimal device characteristics. Here, we report a rationally designed organic macromolecule that simultaneously enables two important advancements to implement ultrasmall organic neuromorphic devices. On one hand, the incorporation of plenty of oxygen-containing moieties in polymer $PBFCL_{10}$ allows significant ion–molecule interaction that fine-tunes the macromolecule's electrical behavior via ion migration and conductive filament (CF) evolution. On the other hand, the densely oriented nanoscale polymeric grains formed with the strong π-π ordering of the rigid furan chromophores regulate the size, interspacing, and spatial arrangement of the metal nanofilaments effectively. By shrinking the diameters and separations of the CFs formed along the oriented polymer grains to nanometer scale in $PBFCL_{10}$ thin film, we are able to fabricate organic synapses with the current smallest size of 50 nm and largest integration dimension of 32 × 32 in compact neuromatrix. Each such device contains a single Ag conductive nanofilament, fine-tuning of which results in the linear evolution to 32 conductance states with high switching uniformity approaching 99%. A mixed-signal neuromorphic hardware system based on the $PBFCL_{10}$ synapse array and an FPGA controller, which are capable of physically executing spiking-plasticity-enabled algorithms and decisive tasks, are implemented to demonstrate the feasibility of utilizing organic synaptic devices for neuromorphic computing.

## Results

### Design of high-performance organic neuromorphic material

Ion migration and conductive filament formation in memristive synapses, which account for the switching of device conductances, occur along the grain boundaries of the organic switching media. It resembles the transport of neurotransmitters and ionic species through the nanoscale cross-membrane channels in biological synapses for neural signal manipulation. The stochastic nature of ion migration and conductive filament evolution in the structurally disordered organic materials, nevertheless, usually leads to the production of synaptic devices with poor uniformity and reliability (Supplementary Fig. 1), making dimensionality shrinking to the nanometer scale and integration into high-density matrix difficult for practical applications. In case that the morphology of the organic switching media can be controlled with more ordered molecular packing, thin film crystallinity, and grain

boundary alignments, the homogeneity of the CF formation and device electrical characteristics will be effectively enhanced. A similar strategy was developed based on dislocation engineering to regulate the switching reproducibility in SiGe epitaxial memory devices[33]. To achieve this target, we design a crystalline control strategy to homogenize the formation of metal conductive nanofilaments in a biomass-based organic switching medium. By using structurally ordered organic materials to achieve uniform distribution of the CFs, the electrical behavior variations of the miniaturized device fabricated on different areas of the switching medium will be greatly suppressed.

In this work, a semicrystalline macromolecule poly(butylene furandicarboxylate)$_{90}$-$b$-($\varepsilon$-caprolactone)$_{10}$ ($PBFCL_{10}$) containing a large number of oxygen moieties was synthesized as the switching matrix for ion-based organic memristive synapse (Fig. 1a and Sections 2-3 in Supplementary Information). The butylene furandicarboxylate (BF) segments with rigid furan ring can provide the necessary molecular crystallinity to form a structure ordered thin film[34], while the incorporation of small portion linear flexible $\varepsilon$-caprolactone (CL) components will enhance both solution processability and mechanical flexibility of the material. The moderate wriggle of the flexible CL segment also facilitates the migration of metal cations through their association and disassociation with the oxygen moieties, offering the operating principle for artificial synapses. We fabricate a 150 nm thin film of $PBFCL_{10}$ on Ag/SiO$_2$/Si substrate by spin-coating (Supplementary Fig. 7). Due to the presence of rigid furan groups in the chemical structure, ordered molecular stacking through π-π interaction between the aromatic moieties can be achieved to form organic crystallites. As plotted in the X-ray diffractive spectrum of Supplementary Fig. 8a, the appearance of the scattering peaks with the $2\theta$ angles of 18.3° and 21.3° can be ascribed to the (001) and (010) crystal faces of the lamellar crystals composed of the furan segments[35], respectively. It corresponds to the unit cell of the furan segment shown in Fig. 1a. During spin-coating, the great centrifugal tendency of the high-speed spinning samples will provide the strong driving force for the $PBFCL_{10}$ crystallites to slide agilely on the substrate, which in turn results in the in-plane vortex alignment of the lamellar crystals (001) in the polymer layer as confirmed by the isotropic scattering ring of the two-dimensional GIWAXS image (Supplementary Fig. S8b and Fig. 1b). Zooming-in observation reveals that in microscopic region the lamellar crystals are all aligned along the radial direction of the spinning sample. The existence of these densely oriented crystals will benefit the formation of an ordered network for metal ion migration along the polymer grain boundaries (left plane of Fig. 1c). In the meanwhile, the strong π–π interactions between the rigid furan segments and thermal annealing at 60 °C will release the internal strain accumulated during thin film formation and promote the vertical stacking and ordered alignment of the (010) lattices in the out-of-plane direction (right plane of Fig. 1c).

In good agreement with the phenomenological model derived from the GIWAXS analysis, atomic force microscopic observation shows that the $PBFCL_{10}$ thin film exhibits smooth surface morphology with the root-mean-square (rms) roughness of 0.37 nm and orderly aligned fibrillar grains with the diameter of ~45 nm (Fig. 1d). Conductive-atomic force microscopy (C-AFM) scanning over a 1 µm$^2$ area of the ON state $PBFCL_{10}$ layer reveals a dense distribution of local high-current regions and thus conductive filaments in the right panel of Fig. 1e, suggesting that quasi-bulk memristive switching occurs in the polymer sample (Section 4 of Supplementary Information). In accordance with the hypothesis that ion migration and CF formation occur along the grain boundaries, as well as the 45 nm diameter of the polymer fibrillar lamellae, the inter-spacing between the neighboring CFs is ~40 nm to 50 nm. Further scanning over a 50 nm × 50 nm area indicates that the largest diameter of the local high-current region in the ON state $PBFCL_{10}$ layer is ~40 nm (left panel of Fig. 1e). Note that the operation methods in C-AFM and device measurements are different. C-AFM measurements are conducted in a point-to-point scanning manner with the as-obtained

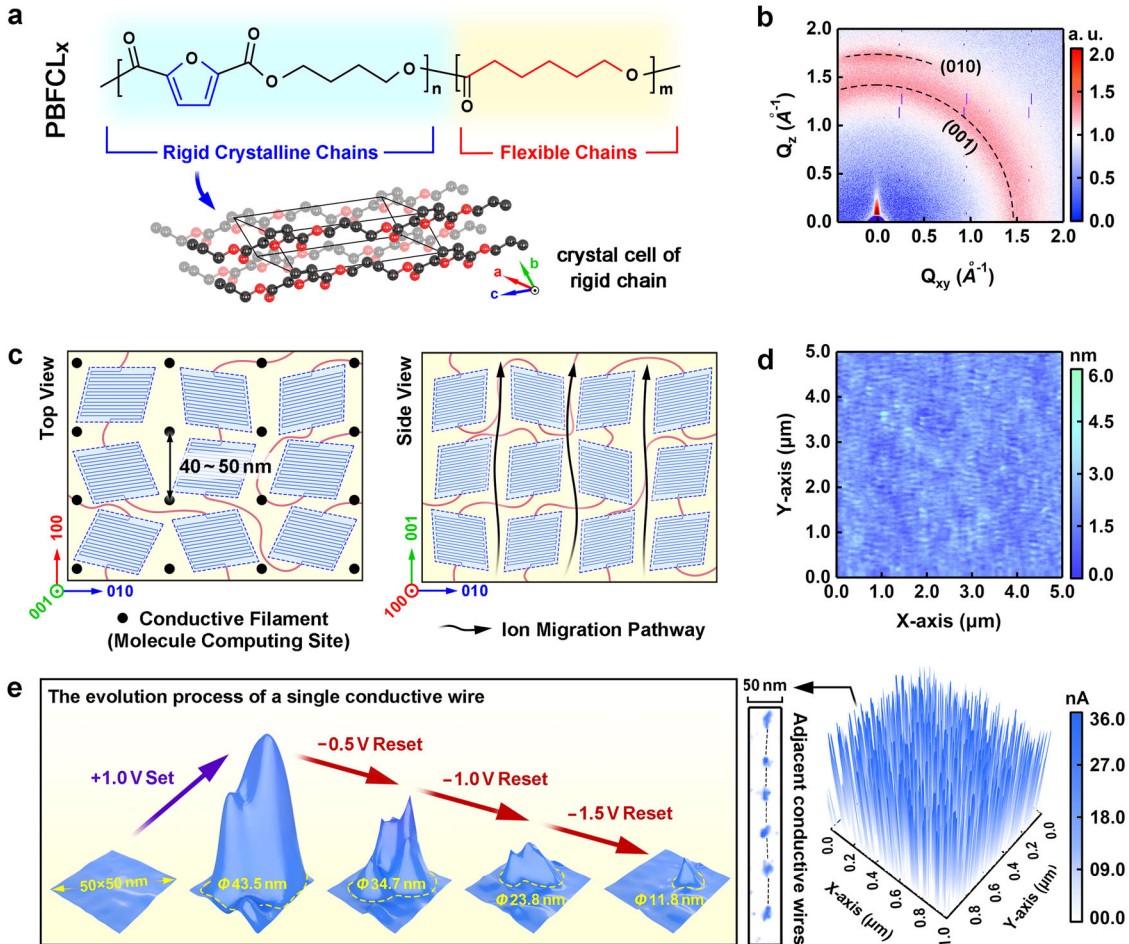

**Fig. 1 | Material design for high-reliability organic neuromorphic matrix.**
**a** Molecular structure, crystalline unit cell, and **b** thin film 2D GIWAXS pattern of
PBFCL$_{10}$. **c** Schematic illustration of orderly aligned nanocrystals and uniform ion
migration paths in PBFCL$_{10}$ thin films (black dot and black curved arrows in the left
and right panels, respectively). **d** Morphology and **e** local ON state conduction
patterns of the PBFCL$_{10}$ thin film. The left panel of **e** shows the evolution of the

conduction region observed in a 50 nm × 50 nm area of the PBFCL$_{10}$ film in the
negative feedback reset process. Local conduction of the polymer was monitored
using C-AFM on a PBFCL$_{10}$ thin film deposited on a silver-coated Si substrate. The
C-AFM tip serves as a top scannable electrode, while the silver coating layer on the Si
substrate acts as both a universal bottom electrode to introduce biased voltages to
the polymer layer and a metal ion source for Ag$^+$ cation injection and CF formation.

current profile showing electrical behaviors of all sampling points in the
scanned area. These high-current regions cover both the fibrillar
lamellae and grain boundaries. In memristive devices, the voltages are
applied through electrodes in one-shot, with the conductive filaments
being formed in areas with the smallest activation energy for ion
migration, e.g., grain boundaries. Considering the dense packing of the
fibrillar lamellae in the PBFCL$_{10}$ layer, the actual size of conductive
filaments in devices should be a few nanometers and much smaller than
the diameter of the high-current regions monitored with C-AFM mea-
surements. These numbers together define the shrinking and separa-
tion extremes of the neuromorphic devices constructed with the
PBFCL$_{10}$ matrix, making downscaling fabrication of nanometer-scale
organic synapses and their compact integration into high-density
crossbar arrays possible. In order to assess the structural stability of the
semicrystalline polymer layer against conductive filament formation-
rupture-regeneration cycles, we reconducted GIWAXS measurements
to monitor the change of scattering patterns of the PBFCL$_{10}$ thin film
after SET and RESET operations. As shown in Supplementary Fig. 9,
although the formation of the nanofilaments between the crystalline
regions may expand the grain boundaries slightly, an almost identical
GIWAXS pattern was observed after the formation and rupture of the
CFs inside the polymer layer. It indicates that the crystallinity and
orientations of the polymer crystals do not vary during and after

filament formation, allowing repeated operation of the organic material
for potentially practical device applications.

When the size of the organic device decreases to 50 nm, a single
complete CF will be formed in each cell along the polymer grain
boundary (Supplementary Fig. 10), avoiding the stochastic evolution
of multiple filaments and therefore suppressing the variation in the
switching parameters. Moreover, the elaborate tuning of the nanofi-
lament with a negative feedback mechanism in the reset process can
lead to the controlled formation of the atomic point contact (APC)
structure (left panel of Fig. 1e and Supplementary Fig. 11)[36], wherein the
decreasing device current would limit the rate of atomic reconfigura-
tion and therefore reduce the APC size in a stepwise manner. As a
result, the APC dimension-related multilevel conductance quantiza-
tion characteristics of the devices may provide an intrinsic linear
conductance modulation platform for precise synaptic weight updat-
ing in neuromorphic computing applications.

## Nanoscale organic neuromorphic device and compact crossbar array
Based on the structural uniformity and quasi-bulk switching char-
acteristics of the PBFCL$_{10}$ thin film, we fabricate a 32 × 32 compact
crossbar array of ultrasmall organic synaptic devices with the current
thinnest linewidth of 50 nm and separation of 85 nm, and the highest

matrix size of 1 Kb (Fig. 2a)[12,30,31,37–59]. Herein, a continuous polymer layer of $PBFCL_{10}$ is used without further patterning. The $Au/PBFCL_{10}/Ag$ devices are thus defined by the inter-crossing top and bottom electrodes. As discussed in the previous section (Supplementary Figs. 12–14 and Fig. 2b), the injection of Ag ions from the silver bottom electrode, their migration across the isotropic polycrystalline polymer matrix, and solid-state redox reactions lead to the formation, rupture, and regeneration of single conductive filament and bistable switching characteristics in the 50 nm $Au/PBFCL_{10}/Ag$ device. During direct current (dc) voltage sweeping operations conducted at room temperature, the organic synapse can be switched between the high resistance (OFF) and low resistance (ON) states at ±0.2 V with an ON/OFF ratio of ~37.2. Energy dispersive spectral mode high-resolution TEM observation of the cross-section of the ON-state $PBFCL_{10}$ synapse devices reveals that a single Ag nanofilament is formed in the organic

switching layer (Supplementary Fig. 13). Transient current measurement indicates that conductance switching from 3.02 $G_0$ to 60.50 $G_0$ occurs within a short period of ~ 21 ns (Fig. 2c). Considering the presence of RC delay for the capacitor like device, another 55 ns should be counted for the nominal switching speed of the organic synapse. In case that further approach can be developed to shorten the RC delay time in future work, the switching speed of the organic neuromorphic device can be decreased significantly. Similarly, a 115 ns ON-to-OFF switching speed of the $PBFCL_{10}$ device, also including the RC delay time, can be recorded in the reset process (Supplementary Fig. 15). We further measure the resistive switching behavior of the organic device in the temperature range of 25–85 °C with a ramping step of 5 °C. As shown in Supplementary Fig. 14b, the lineshapes of the $I$–$V$ curves recorded in this temperature range remain almost unchanged, while the variations of the Set/Reset voltages and ON/OFF state currents

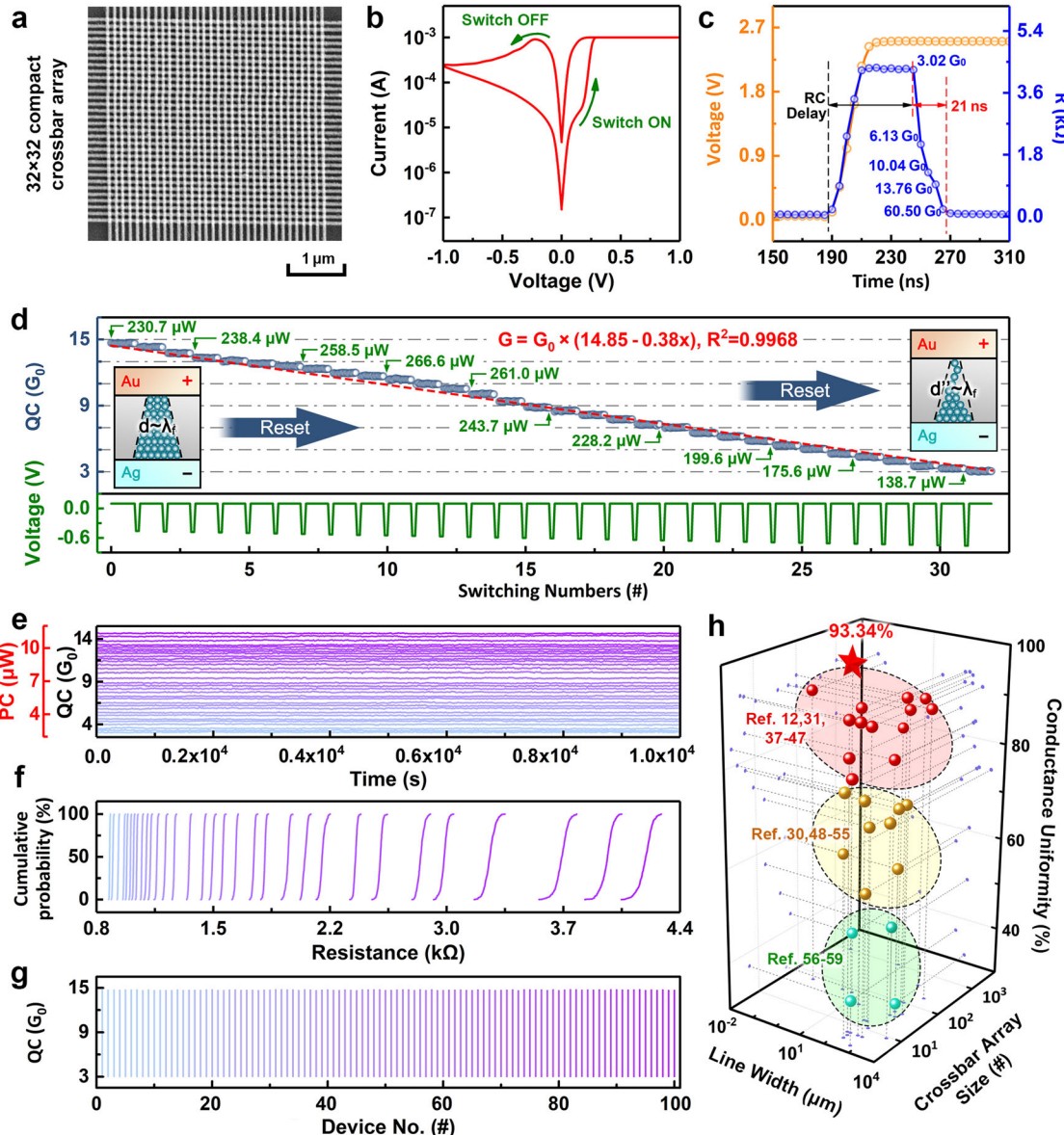

**Fig. 2 | A 32×32 compact crossbar array of 50 nm $Au/PBFCL_{10}/Ag$ devices and its memristive properties. a** Scanning electron microscopy image of the nanoscale $PBFCL_{10}$ synapse crossbar array. **b** Current−voltage characteristics of the $PBFCL_{10}$ device showing bistable switching behavior. **c** Device resistance in immediate response to an applied voltage stimulus. The amplitude of the driving voltage is 2.5 V. **d** Continuous modulation of device conductance in 32 quantized states with respective switching power consumption. **e**−**g** Retention performance with respective reading power consumption, C2C, and D2D statistics of the 32-state quantized conductances. **h** Comparison of the device linewidth, crossbar array size, and switching uniformity of the $PBFCL_{10}$ neuromorphic matrix and other state-of-the-art organic devices as reported in refs. 12,30,31,37−59.

read at 0.1 V are 10.36%, 6.80%, 1.14%, and 7.80%, respectively (Section 6 of Supplementary Information). During the endurance test of $10^{10}$ continuous switching cycles and retention test over $10^6$ s, both of which were conducted at 85 °C, the ON/OFF states currents only show minor variations of less than 5.38% (Supplementary Fig. 14c, d). Extrapolation suggests that the ON/OFF ratio of the PBFCL$_{10}$ synapse remains 34.5 after 10 years, therefore endowing the long-time operation capacity for practical applications.

Beneficial from the structure ordering and uniform distribution of the CFs across the polymer switching layer, devices fabricated on different areas of the organic thin film should exhibit theoretically identical electric characteristics. As shown in Supplementary Fig. 14e, testing over the entire synapse array reveals that 824 out of the 1024 PBFCL$_{10}$ devices show repeatable bistable switching behaviors. Another 123 devices exhibit similar electrical performance yet with minor fluctuations in the switching parameters (Supplementary Fig. 14f). This gives rise to a high production yield of $(824 + 123)/1024 \times 100\% \approx 92.48\%$ for the present organic neuromorphic array. The remaining 77 devices, on the other hand, although show fluctuations in the current–voltage (I–V) characteristics, can also switch binarily (Supplementary Fig. 14g). By extracting the switching parameters from the I–V curves of the 947 stable-switching devices shown in Supplementary Fig. 14e, f, narrow distributions of the programming voltages and ON/OFF state currents read at 0.1 V are depicted (Supplementary Fig. 14h, i), with the device-to-device (D2D) variations of 6.13%, 9.95%, 3.25%, and 6.66%, correspondingly. Without a clear structure–property relationship of the switching characteristics as design guidance, advances in downscaling of organic synapses to the nanometer range or their large-scale integration into a compact crossbar array would be technically less possible (Fig. 2h).

In CF-type synaptic devices, mobile ions determine the physical state of the conductive filaments, while electrons are responsible for conduction through the devices. A slight rearrangement of the ions (atoms) inside the polymer matrix and consequent modifications to the electronic structure of the CFs will lead to obvious changes in the device conductances. Precise control of the atomic configuration of the single conductive nanofilament in PBFCL$_{10}$ synapse therefore plays a critical role in continuous modulation of device conductance. Note that the reset process of the PBFCL$_{10}$ synapse exhibits gradual conductance modulation behavior (Fig. 2b), negative feedback mechanism can be enabled to fine-tune the single CF into an atomic point contact. As plotted in Supplementary Fig. 16a, continuous dc voltage sweeping operation, with the stopping voltages deliberately increasing from −0.45 V to −0.76 V with a step of 0.01 V in the reset process, leads to gradual decreases in the device current. By reading the device current at 0.1 V, 32 consecutive and linearly quantized conductance (QC) states are demonstrated in the PBFCL$_{10}$ synapse (Fig. 2d). The device conductance can be switched between 14.6$G_0$ and 3.1$G_0$ repeatedly with the tuning function of $G = G_0 \times (14.85 - 0.38x)$ and linearity of 0.9968, confirming that the APC is formed in the organic device. Herein, $G_0 = 77.5$ μS is the unit value of the QC that is associated with the one-dimensional confinement of charge carriers during the theoretically scattering-free ballistic transport through the APC, while $x$ is the number of adjusting steps[60]. The deviation of the conductance values from half-integer multiples of $G_0$ may be ascribed to the existence of impurities, imperfect geometry of the APC, as well as its interaction with the molecular matrix that influences the ballistic electron transport through the metallic nanoconfinement, which is also recorded in metal point contact and reported in the literature[61]. When voltage pulses with higher amplitude of −4.0 V and −4.15 V are applied to the PBFCL$_{10}$ synapse, further depression of device conductance from 116.5 μS to 18.6 nS is observed (Supplementary Figs. 18 and 19). The nanoampere current of the PBFCL$_{10}$ device during sub-QC stats modulation is even ∼100 times smaller than that of the oxygen

anion (or oxygen vacancy) migration-based valence change memory (VCM) type devices[62].

Note that all the 32 quantized conductance states can be retained stable for at least $10^4$ s at room temperature in a randomly selected PBFCL$_{10}$ synapse, with maximum fluctuation of 1.60% after the retention test (Fig. 2e). These 32 quantized conductance states can be differentiated clearly without any intersection (Supplementary Fig. 17). The power consumptions of switching and reading (with 0.1 V voltage) these quantized conductance states reach as low as 138.7–266.6 μW and 2.35–11.36 μW, respectively, which is sufficient for synaptic weight updating and VMM operations in low power neuromorphic computing applications. Moreover, the device conductances are all distributed in a narrow range during continuous switching (Fig. 2f). To evaluate the reliability of the organic synapses, we calculate the cycle-to-cycle (C2C) and D2D uniformities of the 32-state conductance quantization characteristics. By using the device conductances plotted in Fig. 2f, g, as well as their standard deviation and average values shown in Supplementary Fig. 20, the C2C uniformity of a PBFCL$_{10}$ synapse during 100 continuous quantization cycles and D2D uniformity of 100 devices in the 1 Kb crossbar array are estimated as 98.89% and 99.71%, respectively.

## Neuromorphic computing with the organic neuromatrix

The downscaling, compact integration and precise tuning capabilities of PBFCL$_{10}$ devices fulfill the requirements for the construction of organic neuromorphic computing hardware. As the core module of such systems, the major functions of organic neuromorphic matrix include synaptic weight updating and storage via nonvolatile device conductance modulation, as well as analog domain vector–matrix multiplication (VMM) operations through the Ohm's and Kirchhoff's laws. Mathematically, a linear modulation of the device conductances will benefit easy and correct updating of the synaptic weights and guarantee the accuracy of the multiplication operations. Considering that VMM is performed over the entire organic neuromorphic matrix, stable modulation of the device conductances with low D2D variations is the other factor that ensures proper execution of accumulation operation. Physically, in case that large-scale integration of high-yield and miniaturized organic synapses can be made possible, a high-performance neuromorphic computing system with high reliability, large throughput, and low areal cost will be then developed. Fortunately, the present PBFCL$_{10}$-based organic synapse and neuromatrix meet the criteria of these indicators. On the other hand, the rest components of the neural networks, including the pooling, activation, batch normalization, etc. are realized in digital circuits. The updating of synaptic weights in the organic synapses, as well as the execution of VMM on the neuromatrix, are also controlled by the digital circuits through the quartus ii platform. In this contribution, we implement a mixed-signal hardware system mainly composed of an organic synapse matrix and a commercial field programmable gate array (FPGA) for neuromorphic computing applications. As shown in Fig. 3a, the 1 Kb crossbar array of the PBFCL$_{10}$ devices is encapsulated into a 64-pin chip and mounted onto a homemade printed circuit board (PCB) operator. In order to eliminate the sneak path problem of the organic crossbar array, an additional selector layer of PEDOT:PSS/Ag nanoparticle composite is added to form the 1S1R device structure. The threshold switching characteristics of the selector layer allow accurate selection and operation of the target PBFCL$_{10}$ device in the array (Supplementary Fig. 21). Being similar to the devices measured individually on a probe station and a semiconductor parameter analyzer, the conductance of the devices in the encapsulated chip can be tuned linearly and repeatedly with the tuning function of $G = G_0 \times (15.066 - 0.376x)$ (Fig. 3b, c).

For a demonstration of the neuromorphic computing scheme, the PBFCL$_{10}$ synapse-based hardware system was used to solve optimization and decision-making problems. Considering the moderate

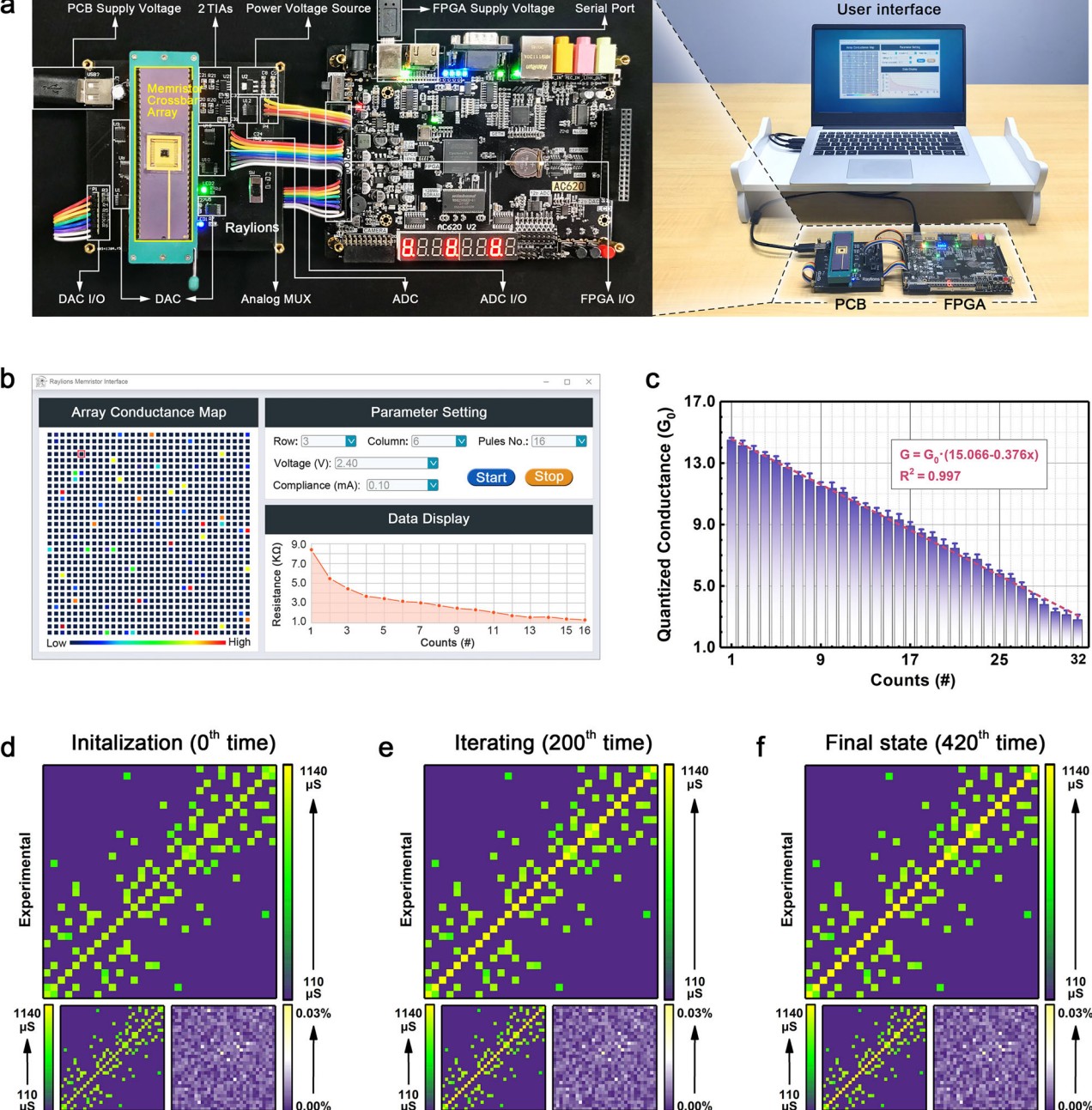

**Fig. 3 | Mixed-signal neuromorphic hardware system based on PBFCL$_{10}$ neuromorphic matrix and digital circuit components. a** Photograph of the hardware system, composed of a PC-based user interface, an FPGA controller, and a PCB operator loaded with a DAC, an ADC, a MUX, TIAs, a power voltage source, an I/O header and a 32 × 32 PBFCL$_{10}$ neuromatrix chip. **b** PC-based user interface for synapse hardware operation. The left panel shows the devices selected for operation and their conductance states in different colors. Upon inputting the column and row numbers of the selected organic neuromorphic devices, program voltages, compliance currents, and numbers of voltage pulses through the user interface, the evolution of the device resistance can be directly modulated

by the FPGA-PCB based system and shown in the bottom right panel of the user interface. The blue rectangle in the upper right panel is the start button to execute the set and reset tasks, respectively, while the orange rectangle is the stop button to arbitrarily end the current task. **c** QC evolution of 100 devices randomly selected in the 32 × 32 neuromorphic matrix chip. *In-operando* updated conductance matrices, theoretical conductance matrices predicted by the synaptic weight updating scheme of BHNN as well as the conductance deviation matrices of the 1024 PBFCL$_{10}$ devices in the 32×32 crossbar array during **d** initialization and after **e** 200 and **f** 420 iterations, respectively.

integration size of the 32 × 32 crossbar array to execute the in-memory computing mode VMM operations, a light-weight Hopfield neural network (HNN) that can be completely projected onto the mixed-signal system is selected for the execution of travel planning tasks (Fig. 4a, b)[63]. Note that optimization and decision-making problems are solved through iteration, traditional learning procedures for

convolution neural networks and pattern recognition tasks are not involved in the present study. The iteration procedure is conducted in situ with the organic neuromorphic hardware. To ensure and accelerate the network convergence into a global minimum, herein we use a biexponential function derived from the spike-rate-dependent plasticity (SRDP) dynamics of the organic device to modify the chaotic

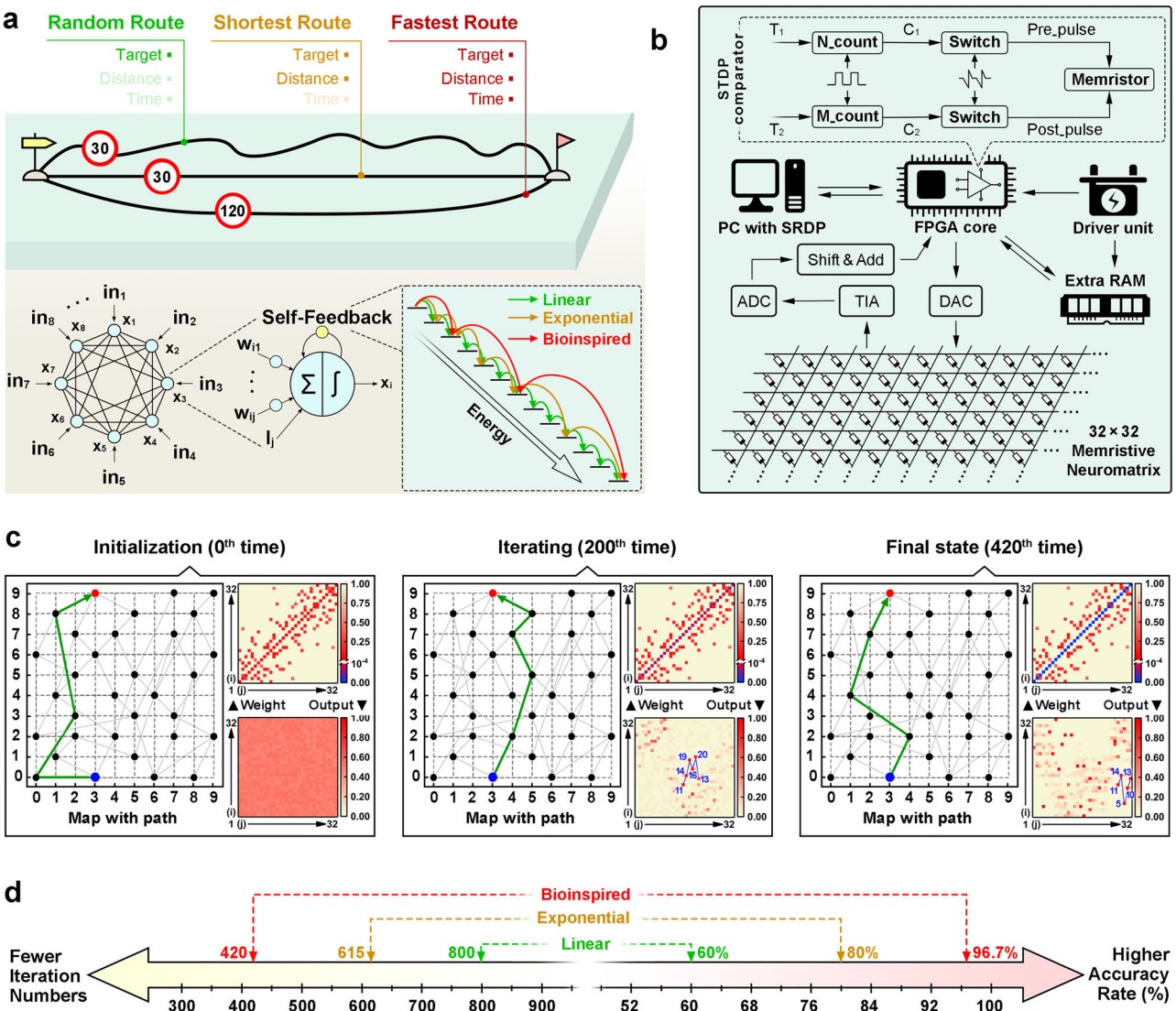

**Fig. 4 | Travel route planning based on the organic neuromorphic hardware and BHNN. a** Upper: schematic illustration of the travel plans with the random, shortest distance and fastest paths. Lower: self-feedback HNNs with the CSA algorithms based on linear, exponential, and biexponential damping factors. The output of a specific neuron originates from the sum of all neurons' inputs and an external bias through an activation function, while the SRDP-based chaos can force and accelerate the system to converge to a global minimum faster with enhanced efficiency. **b** Architecture of the PBFCL$_{10}$ neuromorphic device-based hardware system for the bioinspired HNN, containing a 32 × 32 synapse crossbar array, an FPGA module, an SRDP module for convergence acceleration, an STDP module for temporal comparison, a signal transmission and conversion module between the neuromorphic matrix and the FPGA core, etc. **c** Evolution of the synaptic weight matrices, output matrices and travel routes predicted with the BHNN network during initialization and after 200 and 420 iterations. **d** Comparison of the accuracy rates and numbers of iterations needed to reach the global optimal solution with the linearly-annealed, exponentially-annealed, and bioinspired algorithms.

simulated annealing (CSA) algorithm. A comparator function based on the spike-timing-dependent plasticity (STDP) characteristics of the PBFCL$_{10}$ synapse is also designed to help select the fastest path from the valid candidates found by the network (Sections 9–11 of the Supplementary Information).

Before executing the travel planning tasks, we draw a simplified map of Minhang District of Shanghai Municipal, which includes 32 locations of interests arranged with Arabic numerals in an ordered manner along the longitudes and latitudes (Supplementary Fig. 30). Now assuming that a student is departing from the Hongqiao Railway Station (No. 11) toward the Shanghai Jiao Tong University (No. 13), the major task is to plan an optimal route for him to arrive in no more than 5 virtual minutes. The operating procedure of BHNN is summarized in Supplementary Fig. 31. In this network, the direct distance between locations $i$ and $j$ is normalized and defined as the synaptic weight $w_{ij}$ between neuron $j$ and neuron $i$. As self-feedback, the diagonal weights

of the matrix are arbitrarily initialized as 0.1 (instead of 0, as in the classic HNN model), as shown in the left panel of Fig. 4c. The synaptic weights are physically represented and *in-operando* updated with the conductances of the corresponding PBFCL$_{10}$ devices in the 32 × 32 1S1R matrix (Supplementary Table 4 and Fig. 3d–f). As such, 136 devices in total need to be programmed to low or intermediate resistance states, while the other 888 devices remain in the pristine high resistance state. Initialization of the BHNN also includes inputting the preset time limit ($T_1 = 5$) within which the trip has to be finished, as well as an arbitrary value of 0.01 for the damping factor $\beta$ that controls the annealing rate. The internal and output ($x_i$ and $y_i$) states of the neurons are set randomly during initialization and are iterated according to Section 11 of the Supplementary Information. As the iteration continues, the outputs of the activated neurons increase, while that of the inactivated neurons and the network total energy decrease (middle panel of Fig. 4c and Supplementary Fig. 32a–c). After 420 epochs of

iteration, the global optimal state of the system is found, and the network iteration completes. The final self-feedback weights approach zero, while the outputs of the activated and inactivated neurons become ~ 1 and 0, respectively (right panel of Fig. 4c). A valid and desired route (11-14-5-10-13) with the shortest travel time of 4.166 can be obtained by directly connecting the fully-activated neurons between synapses 11 and 13 along the horizontal axis and reading their respective ordinates as the visiting sequences (Supplementary Fig. 33a–c). As shown in Fig. 3d–f and Section 11 of Supplementary Information, the *in-operando* updated conductance matrix of the synapse crossbar array is in close proximity of 99.77% (1-0.23%) to that predicted by the synaptic weight updating scheme of BHNN, confirming that the organic neuromorphic hardware system is reliable for practical applications.

In order to assess the effectiveness of the spiking-activity-related algorithms on the performance of the organic neuromorphic system, we further compare the iteration efficiencies and accuracies of the classic HNN, linearly-annealed HNN, and BHNN. As shown in Supplementary Fig. 34, the route found by BHNN has the shortest travel time of 4.167 (green line), in comparison to that of the routes found by the initialized network, classic HNN and linearly annealed HNN, estimated as 6.97 (blue line), 6.39 (black line) and 5.27 (red line), respectively. We also compare the SRDP annealing algorithm with the linear and exponential annealing algorithms in terms of the iteration number and searching accuracy (Fig. 4d). Linear annealing costs 800 iteration epochs to reach the global optimum with a low accuracy of 60% in finding the fastest route. Exponential annealing has a higher accuracy of 80% and a more effective convergence in 615 epochs. Both the linear and exponential annealing approaches have limits on their algorithms, as the damping rate cannot be dynamically modulated during iteration. By updating the ramping rate with the SRDP dynamics of the $PBFCL_{10}$ device, the BHNN shows a superior accuracy of 96.7% in less than 420 epochs to reach the global optimal solution. Compared with the exponential annealing, BHNN has a 31.7% $[(615 − 420)/420 × 100\%]$ and a 16.7% (96.7−80%) increase for the computing efficiency and accuracy, respectively, indicating that BHNN is more efficient and accurate for solving optimization problems.

## Discussions

To summarize, we show that by manipulating the crystallinity of the organic macromolecules, a structure-ordered polymer-based neuromatrix with the record high integration of 1 Kb and smallest device size of $50 × 50\ nm^2$ can be achieved with $PBFCL_{10}$ synapses. The construction of a uniform $Ag^+$ cation migration matrix leads to the dense and periodical formation of CFs with diameters much smaller than 50 nm and inter-spacing of ~45 nm, which allows the fabrication of ultrasmall organic neuromorphic devices with a theoretical linewidth limit of 50 nm and separation of 85 nm in $32 × 32$ compact crossbar arrays. After forming process with 5.0 V, Ag conductive filament can be visualized by the EDS mode of TEM. Deliberately tuning of the $Ag^+$ ion migration process enables controlled evolution of single CF and APC in each synaptic device, which in turn gives rise to reliable bistable switching and 32-state linear conductance quantization characteristics with a small conductance variation down to 0.29% and long retention of the QC states $> 10^4$ s, respectively. A mixed-signal hardware system has been built based on the $PBFCL_{10}$ synapse matrix and digital circuits to physically implement the neuromorphic computing paradigm.

It should be noted that the present polymer synapse crossbar array with a device size of 50 nm is constructed with a blanket organic insulator. Further patterning of the switching medium through lithographing and etching on the organic material might result in isolated devices with even better performance. Replacing the foreign mobile ions injected by the electrochemically active metal electrodes with the native ions intrinsically presented in the organic switching materials may also lead to the formation of filaments with a semiconducting nature, which in turn further lowers the device currents and energy consumption for large-scale integration and applications. Nevertheless, care should be paid to balance the paradox between the low-power operation and non-volatile requirements of the devices for data and synaptic weight storage. On the other hand, compared with the present organic-silicon hybrid-integrated neuromorphic system that relies on mathematical algorithms to perform advanced computing tasks, the combination of all flexible organic neuromorphic systems constructed on soft or stretchable substrates and biomimicking short-/long-term plasticities may enable smart bioelectronics that can form dual-way interaction with human beings[10,11,64–68]. Such neuromorphic electronics, which may also be made possible with the $PBFCL_{10}$ synapse devices, show great potential in applications of biomedical implants, augmented reality, and etc[16].

## Methods
### Materials design and synthesis
The design of the organic switching matrix mainly concerns the incorporation of oxygen-containing moieties to provide metal cation migration sites, the crystallinity required to enhance the structural ordering of the switching matrix, as well as the solution processability for device fabrication through spin-coating. In this work, structure-ordered polymer $PBFCL_{10}$ containing rigid furan segments and flexible ε-caprolactone chains were designed and synthesized through polycondensation from poly(ε-caprolactone) (PCL) diol oligomers and biobased 2,5-Furan dicarboxylic acid.

### Device array fabrication and electrical characterization
Nanoscale $Au/PBFCL_{10}/Ag$ devices in $32 × 32$ arrays were fabricated through an electron beam lithography technique and a lift-off approach. The size of the $PBFCL_{10}$ synapse is defined by the line widths of the inter-crossing top and bottom electrode stripes. Electrical measurements of the devices were performed on a Keithley 4200 semiconductor parameter analyzer. For the implementation of an organic neuromorphic hardware system, the synapse crossbar array was connected to the package shell with Al wires through wire-bonding, encapsulated with a $1.9\ cm × 1.9\ cm$ quartz plate into 64-pin chips, and launched on the PCB–FPGA control system. A commercially available field-programmable gate array (FPGA, Altera Cyclone IV model) and a homemade PCB launcher were used in this work (Sections 5 and 8 of the Supplementary Information).

## Data availability
The authors declare that the main data supporting the findings of this study are available within the article and its Supplementary Information files. Extra data are available from the corresponding author upon request.

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

## Acknowledgements

The authors acknowledge the financial support from the National Natural Science Foundation of China (62111540271, 61974090, U19A2053) and the Youth Innovation Promotion Association of the Chinese Academy of Sciences (2018338). The authors also thank Prof. Xiulan Cheng, Ms. Mr. Jian Xu, Fengdan Wang, and Mr. Xuecheng Fu from the Center for Advanced Electronic Materials and Device (AEMD), Shanghai Jiao Tong University, for preparing the organic synapse crossbar arrays.

## Author contributions

S.Z.L., R.Y.Z., Z.Z., P.Z., and G.L. conceived the idea. H.H., R.Y.Z., Y.Z., and R.W. synthesized and characterized the materials. S.Z.L., W.L.C., and J.Z. fabricated the organic synapse crossbar arrays and conducted the electrical measurements. J.Z., A.X., C.L., T.H.H.K., S.J.B., and Z.Z. carried out the mixed-signal neuromorphic system design and tests. Z.X.W., Q.L.C., and P.G. designed the spiking-activity-enhanced HNN algorithms. S.Z.L., H.H., Z.X.W., A.X., X.H., Z.Y., R.Z., Z.Z., P.Z., and G.L. cowrote the paper. All the authors discussed the results and commented on the paper.

## Competing interests

The authors declare no competing interests.
