## [Peer Review File · Nature Communications]

REVIEWER COMMENTS

Reviewer #1 (Remarks to the Author):

This is a very well presented paper demonstrating some substantial improvements in neuromorphic computing architectures using organic materials. My major concern about it is that the concepts have been presented before with other materials and the main innovation presented here concerns fabrication and testing. The organic material and its processing techniques are not completely new either. As far as I can see, there is no fundamental new science and thus the paper must be considered incremental. I therefore think that it is not suitable for Nature Communications and should be submitted to a less general, more specialized journal.

Reviewer #2 (Remarks to the Author):

Shuzhi Liu et al. present a study on a memristive crossbar array with a 50 nm critical dimension and its possible application for Hopfield neural networks construction. The presented new polymer PBFCL10 based memristive device demonstrate fairly good characteristics: long retention time ($>10^6$ s), high endurance ($>10^{10}$ cycles), multilevel switching (32 different states) and low c2c and d2d variation. Moreover, demonstrated 32x32 crossbar array show high yield of working devices ($>92\%$). Overall the presented results are sound, obtained with appropriate techniques, carefully analyzed and could be important to the field of neuromorphic computing on memristors. However, some parts should be revised. The specific concerns are given below (not in the order of importance).

1. The authors are invited to revise the title of the manuscript as actually there was a neuromorphic computing (HNN) demonstration, but not a molecular computing demonstration (memristors do not operate in a single molecular regime).
2. The presented approach to minimize the c2c and d2d variations by controlling a single filament is interesting. Some other approaches for conductive bridge memories could also be discussed (e.g. dislocation engineering (<https://doi.org/10.1038/s41563-017-0001-5>), etc.)
3. Switching time estimations (Fig. 2c and corresponding text) could be reconsidered. Due to capacitor like structure of memristors the RC delay will be always present. So I believe that this time should also be included in the switching time. And what about the switching time of the reset process?
4. Additionally to the previous comment. It is valuable to estimate the energy of switching. The exact values of resistance could not be seen from the Fig. 2c, so my estimations could be wrong (please,

correct if so): $E = U^2 * G * t = 2.5^2 * (3+60) / 2 * 7.75 * 10^{-5} * 20 * 10^{-9} = 300 \text{ pJ}$. This value is good enough for low power neuromorphic computing, but CBM memristors demonstrate 10 fJ range (see e.g. <https://doi.org/10.1002/sml.202006662>). Please, discuss this moment.

5. The quality of the Fig. S11 should be increased. E.g. from the corresponding text I understood that Fig. 11e should present more than 800 different IV-curves, but I can see only one, the same with Fig. S11b – I can see only 2 curves instead of 13.

6. From the Fig. 2e it is difficult to see whether the conductive states intersect or not. Could this figure be replotted (e.g. in supplementary) as distributions of G?

7. SRDP curve (Fig. S20) looks not usual. Usually depending on the post synaptic neuron rate value the synaptic weight decreases and increases (see e.g. in biological synapses (10.1073/pnas.89.10.4363) and in memristive synapses (<https://doi.org/10.1038/s41467-020-15158-3> , <https://doi.org/10.1016/j.neunet.2020.11.005>)). The presented curve demonstrates only potentiation without depression. At the same time depression is also important for learning procedures. Does it possible to get the depression regime in SRDP for your devices?

8. The authors are asked to describe the learning procedure of the BHNN in more details. It is not clear enough how the mathematical model from Section S11 was implemented in hardware. Was the learning implemented ex-situ or in-situ? How many memristors were used?

Reviewer #3 (Remarks to the Author):

Comments to Authors

In the manuscript titled “An ultrasmall organic neuromorphic device for molecular computing”, The authors demonstrated neuromorphic computing hardware based on reliable and high-density organic memristors array by designing the nanostructure of a polymer matrix. Therefore, the reviewer would recommend the publication of this paper to Nature Communications after the following minor revision.

1. The reviewer thinks that the author fabricated a 50nm organic memristor using a new matrix polymer and this topic is not directly related to molecular computing that uses intrinsic properties of specific molecules for demonstrating logic devices, memory devices, etc. In particular, the polymer matrix in the organic memristor may have a 50nm grain, so it is a different structure from typical molecular computing using molecules.

2. In the introduction chapter, please explain the advantages of neuromorphic devices using organic and nanomaterials with the following references. [Joule, 5, 4, 794-810, (2021)/ Nature Electronics 1, 386–397 (2018)/ Nature Nanotechnology 15, 517–528 (2020)]

3. On page 3, the manuscript is started with the advantages of smart bioelectronics. However, it is one of the applications of neuromorphic electronics and this concept is not demonstrated in this research. The author needs to explain the difference between neuromorphic computing and neuromorphic bioelectronics, which rely on long-term synaptic plasticity and short-term, respectively. [Review : Nature Reviews Materials 7, 575–591 (2022)] [Long-term : Nature Communications 13, 4040 (2022)/ Nature Materials 16, 1216–1224 (2017)/ Nature Materials 16, 414–418 (2017)] [Short-term : Nature Electronics 5, 586–595 (2022)/ Science, 360, 6392, 998-1003 (2018)/ Nature Biomedical Engineering, 7, 511–519 (2023)] in the introduction. Additionally, the reviewer recommends showing the additional applications based on short-term memory properties or emphasizing the importance of neuromorphic computing with organic neuromorphic devices versus neuromorphic bioelectronics.

4. In Figure 1, please show a cross-section TEM or SEM image to confirm the formation of a single Ag filament in the 50nmX50nm range. Additionally, please explain the resolution of conductive atomic AFM.

5. In Figure 2, How to ensure that a single 50nm device is composed of single grains after fabrication?

6. The author explained that filament formation occurs between the crystalline regions. Please explain the structural stability of the crystalline region (such as orientation, crystallinity, etc.) during and after filament formation.

7. On page 12, please explain the requirements for the construction of organic neuromorphic computing hardware.

8. In Figure 2, please show quantitative data on retention performances and power consumption.

9. In Figure2, please evaluate the linearity of the organic memristors.

List of Response to Reviewers' Comments and Revisions

Title: “An Ultrasmall Organic Synapse for Neuromorphic Computing”

Authors: Shuzhi Liu, Han Hu, Zhixin Wu, Jianmin Zeng, Ao Xu, Xiaohe Huang, Weilin Chen, Qilai Chen, Zhe Yu, Yinyu Zhao, Rong Wang, Tingting Han, Chao Li, Pingqi Gao, Hyunwoo Kim, Seung Jae Baik, Ruoyu Zhang, Zhang Zhang, Peng Zhou and Gang Liu

ALL THE CHANGES ARE MADE IN RED IN THE REVISED MANUSCRIPT

Response to Reviewers #1' Comments

General Comment:

This is a very well presented paper demonstrating some substantial improvements in neuromorphic computing architectures using organic materials. My major concern about it is that the concepts have been presented before with other materials and the main innovation presented here concerns fabrication and testing. The organic material and its processing techniques are not completely new either. As far as I can see, there is no fundamental new science and thus the paper must be considered incremental. I therefore think that it is not suitable for Nature Communications and should be submitted to a less general, more specialized journal.

Response:

Firstly, we would like to thank the reviewer for recognizing the substantial improvements that we have made for organic neuromorphic computing devices and architectures in this work.

As commented by the reviewer, there is plenty of work reporting organic neuromorphic devices fabricated through spin-coat processing techniques. We are fully aware of this progress, as we have been a major part of it since 2006. However, these works are only conceptual demonstration of organic neuromorphic devices. Their practical applicability has not been proved yet. For instance,

(1) The poor structural uniformity of polycrystalline organic thin films usually results in random distribution of defects and grain boundaries. The uncontrolled migration of mobile ions along these regions consequently lead to stochastic formation and evolution of branched shape conductive filaments. The rupture and regeneration of such branched shape conductive filaments may occur at different locations inside the organic switching layer, giving rise to significant fluctuation of the electrical behaviors of organic neuromorphic devices. This phenomenon is especially severe during device scaling-down.

(2) Organic materials also show poor compatibility with the state-of-the-art CMOS fabrication technology. When processed with these CMOS tools to fabricate neuromorphic device with miniaturized dimension, the involved treatment with high energy electron or optical beams, various acid/base, solvents etc. may damage the structural and functional integrity of the organic switching media seriously.

As a result, the shrinking of organic neuromorphic devices into nanometer range,

as well as their integration into large scale chips, are still difficult at the moment. This severely hinders future development and application of organic neuromorphic computing devices and techniques.

In order to solve these problems, in this work we carefully considered the design of high-performance organic neuromorphic devices from both the fundamental science and technical point of view as following,

(1) From the molecular science aspect, we rationally design the chemical structure of organic polymers to render capability of facilitating ion migration, ordered structure to regulate controlled formation and evolution of conductive filaments, as well as integrity tolerance to the CMOS based fabrication technology. Herein, a novel semicrystalline macromolecule poly(butylene furandicarboxylate)₉₀-*b*-(ϵ -caprolactone)₁₀ (PBFCL₁₀) containing large number of oxygen moieties was designed and synthesized as the switching matrix for ion-based organic memristor. On one hand, the incorporation of small portion linear flexible ϵ -caprolactone (CL) components will enhance both solution processability and mechanical flexibility of the material. On the other hand, the moderate wriggle of the flexible CL segments can facilitate the migration of metal cations through their association and disassociation with the oxygen moieties, offering the conductive filament based working mechanism for artificial synapses. More important, the butylene furandicarboxylate (BF) segments of rigid furan rings can provide necessary molecular crystallinity to form a structure ordered thin film, resulting in dense and uniform formation of conductive nanofilaments. These factors will guarantee stable and reliable electrical performance of the organic synapses. Furthermore, the designed polymer PBFCL₁₀ also shows high tolerance to the CMOS based fabrication process, without any deterioration of thin film quality during electron beam lithography related operations. As such, the fabrication of high-performance organic synapses with miniaturized dimension is made possible through the above rational molecular design strategy.

(2) For the technical part of innovation, we further optimized the fabrication process of organic neuromorphic devices based on electron-beam lithography and related techniques. When fabricating nanometer scale device and high-density cross-bar arrays, the exact geometry of the device that is defined by the perpendicularly intercrossing electrode lines is usually distorted by the proximity effect of E-beam exposure and weakened electron scattering inside the organic layer (in comparison to that in the Si layer). In order to solve this problem, we conducted large amounts of experiments to correlate the E-beam exposure dose and pre-designed electrode pattern with the final geometry of the fabricated devices. With rationally compensated E-beam exposure dose and corrected electrode patterns, we successfully fabricate PBFCL₁₀ neuromorphic devices with the exact pre-defined structure, smallest cell size of 50 nm and largest integration size of 1Kb reported so far.

(3) For the functionality demonstration part of innovation, we finally implement a mixed-signal neuromorphic hardware system through hybrid-integration of the present organic synapse chip and Si-based IC system. In order to achieve this target, we encapsulated the 1Kb array of the PBFCL₁₀ devices into a 64-pin chip and mounted it onto a home-made printed circuit board (PCB) operator. Then we further

modified the algorithms of the Hopfield neural network with the conductance modulating and SRDP dynamics of the PBFCL₁₀ devices, which can also be considered as scientific part of computer science. With the assistance of a commercial field programmable gate array (FPGA) panel, optimization and decision-making task of travel path planning was demonstrated to confirm the potential of organic neuromorphic devices for practical applications.

To the best of our knowledge, this is the world's first hardware implementation of sub-100 nm organic neuromorphic device and Kb level demonstration of organic neuromorphic system towards potentially practical applications. It is a complete, cross-discipline work of chemistry, material science and electrical engineering. None previous reports were able to make such achievements. Instead of being considered as incremental, our innovative designs and discoveries hold substantial importance for the progression of organic neuromatrix for future high-performance computing applications. We believe that the present valuable and insightful findings deserve being published on *Nature Communications*.

Response to Reviewers #2' Comments

General Comment:

Shuzhi Liu et al. present a study on a memristive crossbar array with a 50 nm critical dimension and its possible application for Hopfield neural networks construction. The presented new polymer PBFCL₁₀ based memristive device demonstrate fairly good characteristics: long retention time ($>10^6$ s), high endurance ($>10^{10}$ cycles), multi-level switching (32 different states) and low c2c and d2d variation. Moreover, demonstrated 32x32 crossbar array show high yield of working devices ($>92\%$). Overall the presented results are sound, obtained with appropriate techniques, carefully analyzed and could be important to the field of neuromorphic computing on memristors. However, some parts should be revised. The specific concerns are given below (not in the order of importance).

Response:

We sincerely appreciate the reviewer for his/her positive comments on our manuscript. We have carefully considered the reviewer's comments and revised the manuscript accordingly as described below.

Comment #1:

The authors are invited to revise the title of the manuscript as actually there was a neuromorphic computing (HNN) demonstration, but not a molecular computing demonstration (memristors do not operate in a single molecular regime).

Response:

We thank the reviewer for the suggestion. The title of the manuscript has been revised as "An ultrasmall organic synapse for neuromorphic computing". The statements related to molecular computing have also been revised accordingly throughout the manuscript.

Comment #2:

The presented approach to minimize the c2c and d2d variations by controlling a single filament is interesting. Some other approaches for conductive bridge memories could also be discussed (e.g. dislocation engineering (<https://doi.org/10.1038/s41563-017-0001-5>), etc.)

Response:

We thank the reviewer's praise on our effort in controlling the formation and evolution of a single conductive filament, based on material crystalline structure design and grain boundary engineering to minimize the c2c and d2c variation of the organic synapse device. The strategy of employing dislocation engineering to regulate the switching uniformity of memory devices has been discussed in the revised manuscript on page 5 line 21 to page 6 line 1. The mentioned reference is also cited as Ref. 33 in the revised manuscript.

Comment #3:

Switching time estimations (Fig. 2c and corresponding text) could be reconsidered. Due to capacitor like structure of memristors the RC delay will be always present. So I believe that this time should also be included in the switching time. And what about the switching time of the reset process?

Response:

Supplementary Fig. 15 | Device resistance in immediate response to an applied voltage stimulus of -2.5 V recorded during the reset process.

We thank the reviewer's constructive suggestion and question. Including the existed RC delay time of the capacitor like structure memristor, the nominal OFF-to-ON and ON-to-OFF switching speed were recalculated as 76 ns and 115 ns, respectively. In case that further approach can be developed to shorten the RC delay time in future work, the switching speed of the organic neuromorphic device can be decreased significantly. Related discussion has been revised in the manuscript on page 10 line 4-10, as well as added in the revised Supplementary Information on page 23 line 5 to page 24 line 6.

The above figure displays the transient response of the device resistance to the applied voltage stress of -2.5 V recorded during the reset process.

Comment #4:

Additionally to the previous comment. It is valuable to estimate the energy of switching. The exact values of resistance could not be seen from the Fig. 2c, so my estimations could be wrong (please, correct if so): $E = U^2 * G * t = 2.5^2 * (3+60)/2 * 7.75 * 10^{-5} * 20 * 10^{-9} = 300$ pJ. This value is good enough for low power neuromorphic computing, but CBM memristors demonstrate 10 fJ range (see e.g. <https://doi.org/10.1002/sml.202006662>). Please, discuss this moment.

Response:

We thank the reviewer for his/her suggestion. The energy consumed during switching can be calculated from Fig. 2c according to the equation $E = \int_0^T V_t^2 G_t dt$, where E is the energy consumption, V_t is the voltage applied onto the device at time t while G_t is the device conductance recorded at time t . For the 21 ns switching between 3.02 G_0 and 60.50 G_0 (from 244 ns to 265 ns in Fig. 2c), the energy consumption is 143.27 pJ. Taking the RC delay period into consideration, this number is increased to 186.39 pJ. We agree with the reviewer that such values of the energy consumption are good enough for low power neuromorphic computing.

On the other hand, the memristor device mentioned by the reviewer is based on charge hopping mechanism. Without disrupting and regenerating the conductive filaments, the energy consumption for such devices can be significantly lower. Nevertheless, these kinds of devices usually suffer from short retention problems.

Related discussion was included in the revised Supplementary Information on page 23 line 5 to page 24 line 1. The above-mentioned reference is cited in the revised Supplementary Information as Ref. 7.

Comment #5:

The quality of the Fig. S11 should be increased. E.g. from the corresponding text I understood that Fig. 11e should present more than 800 different IV-curves, but I can see only one, the same with Fig. S11b – I can see only 2 curves instead of 13.

Response:

We thank the reviewer for his/her suggestion. In order to make the multiples I-V curves in Fig. S11b more visible, we use different colors to differentiate them. Nevertheless, as the present PBFCL₁₀ devices show reproducible switching characteristics in the temperature range of 25 degC to 85 degC, it is difficult to differentiate these multiple I-V curves that do not display significant variations. In Fig. S11e, we plot the I-V curve of the first device in red color and those of the remaining 823 devices in grey to differentiate them. Similarly, these I-V curves without significant variations cannot be differentiated clearly. Supplementary Fig. 11 is re-numbered as Supplementary Fig. 14 in this revision.

Supplementary Fig. 14b. Current-voltage characteristics of the Au/PBFCL₁₀/Ag device recorded in the temperature range of 25 °C to 85 °C.

Supplementary Fig. 14e. 824 Au/PBFCL₁₀/Ag devices with repeatable bistable switching characteristics.

Comment #6:

From the Fig. 2e it is difficult to see whether the conductive states intersect or not. Could this figure be replotted (e.g. in supplementary) as distributions of G?

Response:

We thank the reviewer for his/her suggestion. We replotted Fig. 2e as distribution of the device conductance in Supplementary Fig. 17 of the revised SI. It is clearly that the 32 QC states do not intersect with each other. Related discussion is added in the revised manuscript on page 12 line 20-21 and in the revised Supplementary Information on page 30 line 2-3.

Supplementary Fig. 17. The distribution of the 32 QC states with data retrieved from Fig. 2e.

Comment #7:

SRDP curve (Fig. S20) looks not usual. Usually depending on the post synaptic neuron rate value the synaptic weight decreases and increases (see e.g. in biological synapses (10.1073/pnas.89.10.4363) and in memristive synapses (<https://doi.org/10.1038/s41467-020-15158-3>, <https://doi.org/10.1016/j.neunet.2020.11.005>)). The presented curve demonstrates only potentiation without depression. At the same time depression is also important for learning procedures. Does it possible to get the depression regime in SRDP for your devices?

Response:

Supplementary Fig. 25b Spike-rate-dependent plasticity (SRDP) of the PBFCL₁₀ device showing conductance variation with the increase of voltage pulse interval time in the depression processes.

Supplementary Fig. 26. The transient currents of the Au/PBFCL₁₀/Ag device in response to the depression process with the same pulse numbers of 11, amplitude of -2.4 V, width of 10 μs while different intervals of (a) 160 μs , (b) 80 μs , (c) 50 μs , (d) 40 μs , (e) 24 μs and (f) 20 μs , respectively.

We thank the reviewer for his/her question. By employing voltage pulse trains with the amplitude of -2.4 V and intervals varying between 160 μs and 20 μs , SRDP characteristics of the present PBFCL₁₀ devices were also recorded in the depression process, as shown in the above listed Supplementary Fig. 25b and 26. The above-mentioned references are cited in the revised Supplementary Information as Ref. 53-55.

Comment #8:

The authors are asked to describe the learning procedure of the BHNN in more details. It is not clear enough how the mathematical model from Section S11 was implemented in hardware. Was the learning implemented *ex-situ* or *in-situ*? How many memristors were used?

Response:

We thank the reviewer for reminding us to provide a clearer description of the Hopfield neural network. Note that optimization and decision-making problems are solved through iteration of BHNN, traditional learning procedures for convolution neural network and pattern recognition tasks are not involved in the present study. Nevertheless, the iteration procedure is conducted *in-situ* with the organic neuromorphic hardware.

In this work, in order to run the BHNN neural network for optimization and travel path planning task, SRDP and STDP related chaotic simulated annealing (CSA) algorithm (including all the mathematical models of equations 12-18 in the Supplementary Information) is directly burnt into and executed by the FPGA controller through quartus ii platform. During iteration, the synaptic weight updating and storage is achieved *via* nonvolatile conductance modulation of the organic synapses, while the analog domain vector-matrix multiplication (VMM) operations are performed over the entire 32×32 neuromatrix through the Ohm's and Kirchhoff's laws. These are core operations of neuromorphic computing. All the 1024 devices are involved. Among them, 104 non-diagonal devices are programmed to intermediate conductance states according to the design rules and boundary conditions of the neural network and travelling map used for path planning (Fig. 3d). Herein, the synaptic weight w_{ij} between neuron j and neuron i of BHNN is physically represented by the conductance of the organic device located in i^{th} row and j^{th} column of the neuromatrix. These synaptic weights do not vary during the iteration procedure. As self-feedback, the diagonal weights of the matrix are arbitrarily initialized as 0.1 and represented by the low resistance state conductances of the diagonal devices in the neuromatrix. The conductances of these 32 diagonal devices evolves along with the iteration procedure. The other 888 devices are maintained in the pristine high resistance states during iteration. The synaptic weight (device conductance) updating, VMM operations *via* Ohm's and Kirchhoff's laws, and reading of the calculation results are controlled by the FPGA controller. The rest components of the neural networks, including the pooling, activation, batch normalization and etc. are also realized by the FPGA controller.

Related descriptions are included in the revised manuscript on page 13 line 11 to page 14 line 5, page 14 line 22 to page 15 line 3, as well as page 15 line 15-22.

Response to Reviewers #3' Comments

General Comment:

In the manuscript titled "An ultrasmall organic neuromorphic device for molecular computing", The authors demonstrated neuromorphic computing hardware based on reliable and high-density organic memristors array by designing the nanostructure of a polymer matrix. Therefore, the reviewer would recommend the publication of this paper to Nature Communications after the following minor revision.

Response:

We sincerely appreciate the reviewer for his/her positive comments on our manuscript. We have carefully considered the reviewer's comments and revised the manuscript accordingly as described below.

Comment #1:

The reviewer thinks that the author fabricated a 50 nm organic memristor using a new matrix polymer and this topic is not directly related to molecular computing that uses intrinsic properties of specific molecules for demonstrating logic devices, memory devices, etc. In particular, the polymer matrix in the organic memristor may have a 50 nm grain, so it is a different structure from typical molecular computing using molecules.

Response:

We thank the reviewer for the kind reminder. The title of the manuscript has been revised as “An ultrasmall organic synapse for neuromorphic computing”. The statements related to molecular computing have also been revised accordingly throughout the manuscript.

Comment #2:

In the introduction chapter, please explain the advantages of neuromorphic devices using organic and nanomaterials with the following references. [Joule, 5, 4, 794-810, (2021)/Nature Electronics 1, 386–397 (2018)/Nature Nanotechnology 15, 517–528 (2020)].

Response:

We thank the reviewer for the suggestion. Discussing the advantages of neuromorphic devices using organic and nanomaterials will make the presentation of our manuscript logically clearer. Related discussion is included in the revised manuscript on page 3 line 7-11. We also cite the mentioned references as Ref. 13-15 in the revised manuscript.

Comment #3:

On page 3, the manuscript is started with the advantages of smart bioelectronics. However, it is one of the applications of neuromorphic electronics and this concept is not demonstrated in this research. The author needs to explain the difference between neuromorphic computing and neuromorphic bioelectronics, which rely on long-term synaptic plasticity and short-term, respectively. [Review: Nature Reviews Materials 7, 575–591 (2022)] [Long-term: Nature Communications 13, 4040 (2022)/Nature Materials 16, 1216–1224 (2017)/Nature Materials 16, 414–418 (2017)] [Short-term: Nature Electronics 5, 586–595 (2022)/Science, 360, 6392, 998-1003 (2018)/Nature Biomedical Engineering, 7, 511–519 (2023)] in the introduction. Additionally, the reviewer recommends showing the additional applications based on short-term memory properties or emphasizing the importance of neuromorphic computing with organic neuromorphic devices versus neuromorphic bioelectronics.

Response:

We thank the reviewer for reminding us the difference in neuromorphic computing and neuromorphic bioelectronics. In order to avoid misleading, we have removed the description of smart bioelectronics from the introduction section, as the present manuscript is mainly discussing the implementing of neuromorphic computing architecture using organic synapses.

On the other hand, we discussed the importance of developing neuromorphic bioelectronics using organic synapses, as suggested by the reviewer. In comparison to the present organic-silicon hybrid-integrated neuromorphic system that relies on mathematical algorithms to perform advanced computing tasks, the combination of flexible organic neuromorphic system constructed on soft or stretchable substrates and biomimicking short-/long-term plasticities may enable smart bioelectronics that can form dual-way interaction with human beings. Such neuromorphic electronics, which may also be made possible with the PBFCL₁₀ synapse devices, show great potential in applications of biomedical implants, augmented reality, etc.

Related discussion is included in the Discussion section of the revised manuscript on page 18 line 13-20. The above-mentioned references are also cited as Ref. 10, 11, and 46-50 in the manuscript.

Comment #4:

In Figure 1, please show a cross-section TEM or SEM image to confirm the formation of a single Ag filament in the 50nmX50nm range. Additionally, please explain the resolution of conductive atomic AFM.

Response:

We thank the reviewer for the constructive comments. As requested, we have performed high-resolution TEM observation to confirm the formation of a single Ag filament in the 50 nm×50 nm range. To be more representative, instead of using the C-AFM sample without top electrode for TEM observation, we directly prepare a cross-sectional specimen from the ON-state PBFCL₁₀ device in the 32×32 crossbar array using focus-ion beam techniques, to visualize the formation of a single Ag nano filament in the 50 nm×50 nm device area. As indicated by the energy dispersive spectral profile of Supplementary Fig. 13, a weak orange region can be seen dimly inside the polymer layer between the top Au/Ti (red/green) and bottom Ag (orange) electrodes. It indicates that a single Ag filament with the size of 5 nm ~ 10 nm is formed in the device.

Supplementary Fig. 13. Energy dispersive spectral mode cross-sectional TEM image of the ON-state Au/Ti/PBFCL₁₀/Ag/Ti device.

On the other hand, the resolution of the C-AFM measurement is 256 pixel/1 μm , equaling to 3.91 nm/pixel. Therefore, the conductive filament with the size of a few nanometers can be faithfully observed through C-AFM measurements.

Related discussions were included in the revised manuscript on page 9 line 22 to page 10 line 3, as well as in the revised Supplementary Information on page 14 line 11-13 and on page 19 line 18 to page 20 line 4.

Comment #5:

In Figure 2, How to ensure that a single 50 nm device is composed of single grains after fabrication?

Response:

We thank the reviewer for his/her question. In this work, we are not claiming that a 50 nm PBFCL₁₀ device is composed of a single polymer grain. Instead, what the device contains is no more than one grain boundary.

As shown in Fig. 1d, the AFM observation reveals that the diameter of the polymer fibrillar lamellae in the PBFCL₁₀ thin film is ~ 45 nm. According to the hypothesis that ion migration and CF formation occur along the grain boundaries, the smallest inter-spacing between the neighboring CFs is thus ~ 40 nm to 50 nm. As schematically illustrated in Supplementary Fig. 10, when the organic synapse size scales down to 50 nm, no more than one grain boundary will exist in the device. Consequently, only one conductive nanofilament will be formed in the 50 nm size PBFCL₁₀ device, the fine-tuning of which will lead to the formation of atomic point contact structure showing conductance quantization characteristics.

Related discussion is included in the revised Supplementary Information on page 15 line 9 to page 16 line 3.

Supplementary Fig. 10. Schematic illustration of relationship between the size of the conductive filament, the size of the crystalline grains and the 50 nm PBFCL₁₀ device.

Comment #6:

The author explained that filament formation occurs between the crystalline regions. Please explain the structural stability of the crystalline region (such as orientation, crystallinity, etc.) during and after filament formation.

Response:

We thank the reviewer for the valuable suggestion. To evident the structural stability of the crystalline region, we re-conduct GIWAXS measurements after C-AFM based SET and RESET operations on the PBFCL₁₀ film. As displayed in Supplementary Fig. 9 and Fig. 1b, the scattering patterns of the PBFCL₁₀ film show similar crystal assignment and intensity after the formation of conductive filaments. It suggests that although the formation of CFs may expand the grain boundaries slightly, the crystallinity and orientations of the polymer crystals are well maintained. Therefore, the structural stability of the semicrystalline PFBCL₁₀ endows itself great potential for practical neuromorphic device applications.

Related discussions are included in the revised manuscript on page 8 line 13-21 and in the revised Supplementary Information on page 15 line 1-7.

Supplementary Fig. 9. GIWAXS pattern of PBFCL₁₀ film after SET and RESET operations.

Comment #7:

On page 12, please explain the requirements for the construction of organic neuromorphic computing hardware.

Response:

We thank the reviewer for the suggestion. As the core module of organic neuromorphic computing hardware, the major functions of organic neuromorphic matrix include synaptic weight updating and storage *via* nonvolatile device conductance modulation, as well as analog domain vector-matrix multiplication (VMM) operations through the Ohm's and Kirchhoff's laws. Mathematically, a linear modulation of the device conductances will benefit easy and correct updating of the synaptic weights and guarantee the accuracy of the multiplication operations. Considering that VMM is performance over the entire organic neuromorphic matrix, stable modulation of the device conductances with low D2D variations is the other factor that ensure proper execution of accumulation operation. Physically, in case that large scale integration of high-yield and miniaturized organic synapses can be made possible, high-performance neuromorphic computing system with high reliability, large throughput and low areal cost will be then developed. Fortunately, the downscaling, compact integration, and precise tuning

capabilities of the present PBFCL₁₀ based organic synapse and neuromatrix fulfill all these requirements for the construction of organic neuromorphic computing hardwares.

Related discussion is included in the revised manuscript on page 13 line 11 to page 14 line 2.

Comment #8:

In Figure 2, please show quantitative data on retention performances and power consumption.

Response:

We thank the reviewer for the suggestion. As requested, we calculate the fluctuations of the device conductances in the 32 QC states. After 10⁴ s retention performance tests, the maximum fluctuation of the quantized conductances is only 1.60%. It again confirms the stability and reliability of the present PBFCL₁₀ synapse device for long term operation. On the other hand, we also calculate the power consumptions to switch the PBFCL₁₀ devices among the 32 QC states, as well as that to read these states. By multiplying the device conductance with the switching voltages or reading voltages, the power consumptions of switching and reading (with 0.1 V voltage) these quantized conductance states range between 138.7 to 266.6 μW and 2.35 to 11.36 μW, respectively (Fig. 2d and 2e). They are sufficiently low for the synaptic weight updating and VMM operations in neuromorphic computing applications.

Related discussions are included in the revised manuscript on page 12 line 19 to page 13 line 2.

Fig. 2d Continuous modulation of device conductance in 32 quantized states with respective switching power consumption.

Fig. 2e Retention performance of the 32-state quantized conductances with respective reading power consumption.

Comment #9:

In Figure 2, please evaluate the linearity of the organic memristors.

Response:

We thank the reviewer for the suggestion. As stated in the manuscript on page 12 line 4-6, as well as in the Supplementary Information on page 29 line 7-13, the linearity of conductance quantization as shown in Fig. 2d is 0.9968 for the present PBFCL₁₀ memristive synapses. It is calculated according to the following equation,

$$R^2 = [\sum_{i=1}^n (\hat{y}_i - \bar{y})^2] / [\sum_{i=1}^n (y_i - \bar{y})^2] \quad (8)$$

Where R^2 is the linearity of conductance quantization, \hat{y}_i and y_i represent the fitted and experimental value of device conductance, while \bar{y} is the average value of the experimental device conductance.

REVIEWERS' COMMENTS

Reviewer #2 (Remarks to the Author):

The authors have addressed all my comments. I believe the manuscript could be published in its present form.

REVIEWERS' COMMENTS

Reviewer #2 (Remarks to the Author):

The authors have addressed all my comments. I believe the manuscript could be published in its present form.

Response:

We thank Reviewer #2 for his/her constructive evaluation of our work and the explicit recommendation for publication.